# Amplifiers in Biomedical Engineering: A Review from Application Perspectives

**DOI:** 10.3390/s23042277

**Published:** 2023-02-17

**Authors:** Lida Kouhalvandi, Ladislau Matekovits, Ildiko Peter

**Affiliations:** 1Department of Electrical and Electronics Engineering, Dogus University, Istanbul 34775, Turkey; 2Department of Electronics and Telecommunications, Politecnico di Torino, 10129 Turin, Italy; 3Department of Measurements and Optical Electronics, Politehnica University Timisoara, 300006 Timisoara, Romania; 4Istituto di Elettronica e di Ingegneria dell’Informazione e delle Telecomunicazioni, National Research Council, 10129 Turin, Italy; 5Department of Industrial Engineering and Management, University of Medicine, Pharmacy, Science and Technology “George Emil Palade”, 540139 Targu Mures, Romania

**Keywords:** amplifiers, biomedical applications, disease, implantable amplifier designs, industrial, scientific, medical (ISM) bands, patient

## Abstract

Continuous monitoring and treatment of various diseases with biomedical technologies and wearable electronics has become significantly important. The healthcare area is an important, evolving field that, among other things, requires electronic and micro-electromechanical technologies. Designed circuits and smart devices can lead to reduced hospitalization time and hospitals equipped with high-quality equipment. Some of these devices can also be implanted inside the body. Recently, various implanted electronic devices for monitoring and diagnosing diseases have been presented. These instruments require communication links through wireless technologies. In the transmitters of these devices, power amplifiers are the most important components and their performance plays important roles. This paper is devoted to collecting and providing a comprehensive review on the various designed implanted amplifiers for advanced biomedical applications. The reported amplifiers vary with respect to the class/type of amplifier, implemented CMOS technology, frequency band, output power, and the overall efficiency of the designs. The purpose of the authors is to provide a general view of the available solutions, and any researcher can obtain suitable circuit designs that can be selected for their problem by reading this survey.

## 1. Introduction

The need for the healthcare system is increasing, as factors such as life style, food habits, environmental schemes, etc., are altering and may be deteriorating. As a result, healthcare offices and hospitals are required to have advanced equipment to be able to handle the increasing number of requirements of patients [1]. The integrated circuit (IC) technology used within biomedical implants is growing exponentially [2,3,4]. From another perspective, Internet of Things (IoT) systems will be fruitful in emerging applications that are based on radio frequency (RF) designs. IoT systems include computer, Internet, and mobile communication technologies [5]. Low-power RF circuits can be employed in industrial, scientific, and medical (ISM) bands that are suitable for medical applications. Hence, high-performance designs in terms of noise, bandwidth, linearity, etc., are substantially required [6,7]. In the field of medical applications, there is also a growing body of research on the acquisition and decoding of physiological signals [8].

RF power amplifiers (PAs) are widely used in industry, medical imaging, fifth-generation telecommunication, mobile communication networks, biomedical applications, and so on [9,10]. In medical applications, they are one of the most important technologies. For example, electroencephalograms (EEG) have recently been used for brain recording, as it is less expensive and more safe in magnetic fields [11,12]. For illustrating the importance of these medical devices, consider the example of a particle accelerator typically used for cancer diagnosis and prediction of initial treatment. To have effective performance in particles in a cyclotron, it has been important to use RF amplifiers, namely vacuum tube-type and solid state-type amplifiers [13]. Due to the highly infectious nature of the COVID-19 situation, doctors and patients are interested in wireless communication links [14]. Hence, medical devices, especially implanted devices, are substantially required to provide a ‘contact-less’ situation and to improve the link between performance and reliability [15]. By developing technology, implanted devices can serve the lives of many patients and can be used for clinical treatments [16,17,18].

For biomedical applications, it is essential to focus on the roles of patients, doctors, and nurses as well as the suitable placement of implanted devices inside or outside of the body. Additionally, for the implanted devices, some of the various specifications that must be considered are linearity, bandwidth, data rate, consumed power, and so on. In order to reduce the thermal influence in these devices, the heat generation of the designs must be minimized in order reduce any influence on the physiological activities. Furthermore, the overall circuit must have low power consumption [19].

Due to the importance of amplifiers in biomedical devices, this survey is devoted to summarizing the various circuits and technologies used for designing amplifiers and their use in medical applications. The remainder of this paper is as follows. Section 2 presents the biomedical devices and their benefits from an engineering perspective. Section 3 provides the various up-to-date biomedical and implanted amplifier designs that are used for diagnosing disease and treating patients. Section 4 presents a general view of the designed biomedical amplifiers from their layouts through to their fabrication. Because the measurements of these devices are important, Section 5 presents some of the methods employed for testing and measuring these devices. Finally, Section 6 concludes this manuscript.

## 2. Biomedical Devices from an Engineering Perspective

Figure 1 presents the various body parts where the implanted devices can be installed inside the body. Some of them substitute parts of the body, while others are embedded within various human tissues such as skin, fat, muscle, skull, dura, csf, and brain [20]. To illustrate the importance of these medical devices, brain signal acquisition front-end designs that involve amplifiers were presented in [21] (see Figure 2).

For biomedical devices, the PA design is the essential element for transmitting/receiving RF signals, as presented in Figure 3. Overall, the transmitter is expected to achieve a suitable output power with the given efficiency. The PA structure is generally used for driving antennas and dissipates most of the power. Figure 4 presents the general procedure that antennae and amplifiers are provided for transferring data from patient to a medical doctor.

Hence, the structure and output specifications of the PA play important roles [6].

CMOS-based transceivers (TRX) play an important role in biomedical applications [24,25], and they include both receiving and transmitting blocks as frequency synthesizers, voltage-controlled oscillators, amplifiers, and demodulators. For achieving high-performance TRX, each block must have good performance outcomes for generating and addressing high-magnitude pulses and modulations, respectively [6]. Recently, various methods devoted to the design of well-satisfied TRXs have been proposed, and some of the features exploited by researchers in the biomedical field are reported below.

## 3. Biomedical Amplifiers and Various Designs

The common and general bioelectric system includes the sensing, algorithm, and stimulation blocks [26]. Hence, designing and simulating each of these components requires special attention and multidisciplinary experience, as the chips are used for medical applications. This section provides a summary of various amplifiers used in medical applications using wireless technology.

Recording human electrocorticographic (ECoG) signals, especially for brain activity at a high resolution, suffers from a low signal-to-noise ratio and a low spatial resolution. In [27], an analog front-end (AFE) was presented that is appropriate for recording implants, and it includes a neural amplifier. The designed front-end amplifier provides a merged amplifier analog-to-digital converter (ADC) topology and wireless validation circuit. The architecture is employed in 65 nm CMOS technology with a consumed power of 3.2 μW with a common-mode rejection ratio (CMRR) of 77 dB. The layout of the AFE design includes a V/I converter, a current-controlled ring oscillator (ICRO)-based ADC, and a mixed-domain servo loop that is integrated into the recording system on a chip (SoC).

For the use of neural recording, an eight-channel AFE with a power supply rejection ratio (PSRR) of 110 dB and power supplies of 0.35 and 0.7 V was presented in [28]. Figure 5a presents the proposed amplifier for use in an interface SoC (see Figure 5b) where the maximum gain is 54 dB and the input-referred noise is 6.7 μV. The proposed replica biasing scheme provides suitable PSRR performance with stable bias current for the input stage of the low-noise amplifier (LNA). The consumed power is reduced using an averaged local field potential (A-LFP) servo loop where the signal is generated by integrating the four-channel programmable gain amplifier (PGA) outputs.

For treatment of several neural disorders, deep brain stimulation (DBS) was studied in [29] for a rat model using a battery-less mechanism in freely moving rats. The chip presented in that work is able to communicate with eight different brain regions, and the size of the chip was reduced. The microsystem performs transceiving actions through a single coil. The proposed chip includes a chopper-stabilized recording amplifier, an ADC, a multi-channel stimulation circuit, a programmable information hub, forward telemetry circuits, and reverse telemetry circuits. A detailed schematic of the chopper-stabilized LNA is presented in [29], and it adopts a fully differential architecture. The power per recording channel is 8.6 μW, with a stimulation frequency of 60 Hz to 220 Hz. The data rate for transmitter is 2 M bps and that of the receiver is 100 K bps.

Cruz et al. presented a low-intermediate frequency (IF) RF front-end that is appropriate for medical implant communication services [30]. It consists of an LNA and a mixer from 402 MHz to 405 MHz where the LNA achieves a 10 dB gain with a power consumption of 0.94 mW that is employed in the TSMC 0.18 μm CMOS process. The presented circuit with the size of 1.4 mm × 1.2 mm (see more details in [30]) supports 446 kbps quadrature phase-shift keying (QPSK) modulation, which is a high data rate for a such design.

In various neuroscience experiments, neural amplifiers targeting deep brain regions play important roles for monitoring the disease. A high-power efficiency neural amplifier was presented in [31] with a minimum area of 0.051 mm${}^{2}$ that consumes 2.1 μW using the 1 V supply, offering a DC input impedance of 6.7 GΩ. In [31], a proposed neural amplifier was presented where no current flows from the input. In this design, due to the presence of a gain revision block, the buffer does not require charging of the input capacitor.

As described above, it is critical to have circuits with low power consumption and low complexity in biomedical applications. For wireless medical applications, on–off keying (OOK) design is used in [32] as a modulation/demodulation circuit for 2.4 GHz band with a power consumption of 160 μW in the receiver and 0.6 μW in the transmitter. The presented transceiver includes the receiver, which is subdivided into a input matching network, an LNA, a novel single-ton differential envelope detector, a level shifter, cascaded baseband amplifiers, and a hysteresis comparator. Additionally, the transmitter has a bias-stimulating circuit, a current-reuse self-mixing voltage controlled oscillator, and a quadruple-transconductance power amplifier. Due to the data rate of 20 Mbps and the energy efficiency of 80 pJ/bit, the overall transceiver is appropriate for implantable body sensor networks.

Ghanad et al. presented a local temperature monitoring system that is implantable, and it requires an the average power of 29.5 μW [33]. The power dissipation of whole circuit can be decreased using a time-domain sensor readout and an OOK-modulated free-running oscillator. The implantable chip is designed and fabricated with 0.18 *u*m CMOS technology where the sensor type is resistive with two channels. It has a sensitivity within ±220 Ω. In [33], an implantable local temperature chip with 180 nm CMOS technology was presented. It includes various blocks for the implantable unit, and the temperature recording outcomes were tested inside the hot water. The brown adipose tissue (BAT) channel was employed for higher temperatures and normal body tissue was used for lower temperatures, where this channel recorded temperature above 39 °C.

A Class-E PA was presented in [34] for transferring power to medical implants using wideband frequency-shift keying (FSK) modulation with the carrier frequencies of 2 MHz and 4 MHz. The proposed PA can transmit data with a larger data rate to carrier frequency of 33%. As was previously mentioned, the carrier frequency of 2 MHz is twice that of the other carrier (i.e., 4 MHz), which results in wideband modulation and a power transfer efficiency (PTE) of 25%. The simplified PA is presented in [34], where it is able to transmit power with a data rate of 1 Mbps. In [35], another type of amplifier, a Class-D, was presented that works up to 14 MHz with an approximate efficiency of 80% and a maximum output power of 36 W. In [35], the general architecture of the Class-D PA is designed with 180 nm CMOS technology. It is an integrated PA with the ability to be used in wireless applications and medical transmitters.

In [36], a quad-shank CMOS neural probe was presented for the recording of multiple brain regions. For this case, the fully-differential channels are designed with an amplifier and a 14-bit ADC. For this design, no calibration is required and the total power consumption is 36.5 mW.

Hansen et al. presented a monolithic microwave wave integrated circuit (MMIC) where one of the blocks in the transmitter includes PA, and an LNA is used in the receiver [37]. This combination is a harmonic radar system that can be employed in the ISM frequency bands. The PA is employed here for sensing the transmitter signal and the LNA is used for enhancing the gain and sensitivity. The presented design can be used in the medical area where small reflectors need to be precisely determined.

In [38], a printed charge amplifier was presented for measuring on-skin biosignal. The design consists of organic transistors, and the used substrate is inkjet-printed Ag ink. The prepared measurement setup is presented in [38]. The circuit consists of integrated bias, feedback resistors, and a feedback capacitor that can be used for the evaluation of vascular health. The lower cut-off frequency can be varied by changing the value of the feedback resistor. Additionally, the arterial pulse wave signal is improved using the fully printed charge amplifier.

A fully-integrated low-power full-duplex transceiver (FDT) was presented in [39], where receiver and transmitter sections have one antenna, leading to a reduction in the complexity. The general structure of an implantable neural recording device is shown in Figure 6, which includes a 2.4 GHz on-chip receiver with binary phase-shift keying (BPSK) modulation, and TSMC 180 nm CMOS technology is used. The presented chip has a size of 0.8 m${}^{2}$ and provides an efficiency of 41.6% with a regulated power delivery of 25 mW.

Using the 350 nm CMOS process, a trimodal neural interface system-on-chip (SoC) is explained in [40] that includes 16-channel neural recording. A conceptual view of the presented SoC is shown in Figure 7, which is installed on the head of a moving rat. In this chip, the stimulus artifact rejection injects a series of ±500 μA into the model.

In [41], a chronic implantable device simulated with 180 nm CMOS technology was presented. The general structure is presented in [41], and it includes various blocks such as amplifiers, oscillator, ADC blocks, and so on. This design is a ten-channel peripheral nerve electroneurogram (ENG), and it is validated in vivo for acquisition. Each amplifier channel within the chip is digitized with 10-bit resolution.

For implanted devices, the radio frequency identification (RFID) readers can receive the data, and provide suitable power. In [42], a structure for the RFID reader was presented where it included Class-E/Fodd PAs and incorporated a single-segmented antenna in the receiver and transmitter sections, which operate simultaneously. The typical efficiency of the design is 54%. The transceiver architecture is presented in [42], and the implementation of amplifiers is presented in detail in Figure 8.

An integrated circuit to be used for medical implants is presented in [43], and the related structure includes three subsections (RF powering, transmitter, and signal acquisition) where the amplifier is placed in the transmitter block. The proposed structure includes a parallel-in/serial-out circuit (PISO), an injection-locked frequency divider (ILFD) for data transmission, a PA, and converters. These blocks are used for generating RF power, transmitting, and receiving data. The general structures of ILFD and PA are shown in [43]. The overall circuit generates a low-phase-noise carrier at 402 MHz.

Copani et al. presented a CMOS transceiver that can be used for low-power medical implants [44]. This implementation consists of a super-regenerative oscillator, an LNA, a class AB amplifier, and an injection-locked power oscillator. The general structure of the medical implant communication service (MICS) transceiver is depicted in [44]. Detailed structures of the super-regenerative wake-up oscillator (SRO), LNA, and PA are shown in Figure 9.

An arranged cloud health care network is presented in [45], where an antenna is employed for teeth implantation and an oscillator is connected to the antenna without any need for an amplifier. The used sampling amplifier with the antenna is presented in [45], where a practical implementation of the antenna in the mouth of patients results in a −6.78 dBi gain in performance. The overall structure presented in this study is able to monitor the temperature and transmit data wirelessly using cloud health care. The general structure of the healthcare-monitoring network is described in Figure 10.

In another study, a low-energy receiver using the LNA was reported that was designed in 65 nm CMOS technology [47]. This structure results in a noise figure of 8.2 dB and a gain of 49.5 dB.

In medical ultrasound imaging, endoscopic transducers are widely used and studied. A common problem related to this device is the lower signal-to-noise ratio in the image at high-frequency ultrasound waves. To tackle this problem, in [48] a high-sensitivity transducer with an integrated miniature amplifier was presented. The proposed amplifier can reduce the noise results and provide an enhanced potential for visualizing the deeper tissue of the human body. The general structure of the transducer with the micro-amplifier is presented in Figure 11. The achieved noise figure for the frequency band between 0 and 50 MHz demonstrates that the noise is sufficiently small and cannot affect the transducer waveform.

In [25], a brain–computer interface that can be substituted in the ear of human body with reduced noise was presented. The general structure is presented in Figure 12. The presented structure results in a CMRR larger than 95 dB, and a dc offset (EDO) of 350 mV.

Liu et al. presented a wireless sensor–brain–machine interface (SBMI) system for joining the various sensors on the body of a human, restoring data through a tactile force sensor, an electro-goniometer, and fully integrated wireless BMI [24]. The main structure consists of an ADC, a receiver, and a transmitter, where the blocks of ultra-wideband (UWB) wireless transceivers [24] are shown.

A programmable electroencephalography monitoring system-on-a-chip (SoC) was presented in [16], where an ultra-low power TRX was implemented. The programmable SoC was generated using four sub-blocks, and a high-performance radio frequency TRX was one of the blocks. Figure 13 shows the general structure of the TRX, where the bias-stimulating circuit, a current-reuse self-mixing voltage control oscillator, a quadruple-transconductance power amplifier, and a matching circuit are used in the transmitter. In the receiver, a single-end input to differential-end output envelope detector, a level shifter, a baseband amplifier, and a hysteresis comparator are employed.

In [35], a D-class amplifier was designed to improve the functionality. In this study, the architecture of a wireless power link that can be used for medical implants was presented. This PA was designed and simulated at the 180 nm CMOS technology, results in an efficiency greater than 80%, and operates up to 14 MHz with a 30 V supply voltage. The main part of this D-class PA consists of ML (i.e., n-type low-side DMOS transistor) and MH (p-type high-side DMOS transistor) [35].

To improve low-frequency noise performance, current-feedback instrumentation amplifier (CFIA) was presented in [49], which includes a switched capacitor (SC) integrator for providing a high-pass filter, resulting in offset reduction. This design was simulated in 130 nm CMOS technology with a 1 V supply voltage, 2.3 μA current, and 125 dB CMRR specification. The general structure of the current feedback instrumentation amplifier is presented in Figure 14.

A general method for transferring power to medical implants is an inductive link that is driven by a Class-E PA. In [34], the simplified structure of an inductive link driven by a Class-E PA is depicted. One of the important schemes for data modulation is FSK, and the general structure of FSK data modulation is shown in Figure 15. This presented structure consists of a 7.1 μH transmitter coil with a 1.2 μH receiver coil. The actual measured PTE with power delivered to the load (PDL) specifications are 25% and 126 mW.

The block diagram of a neural recording system is presented in [50]. For this block, a suitable amplifier is designed that can be used for neural recording systems (see Figure 16). This amplifier is appropriate for bio-potential recording.

The front-end amplifier (FEA) is another useful type of amplifier that can be used for detecting biosignals. For high-density implantable applications, an input impedance booster with current compensation feedback technique was presented in [51]. The presented amplifier was designed and simulated in 180 nm CMOS technology. The simulated results revealed an input impedance of 60 GΩ and a CMRR of 61 dB. The operational transconductance amplifier (OTA) circuit is presented in Figure 17.

Luo et al. presented a low total harmonic distortion (THD) chopper amplifier in [52] that is aimed at having a low level of noise. This amplifier was designed and simulated in 180 nm CMOS technology and has a power consumption of 3.24 μW/channel, where the supply voltage is 1.8 V. The detailed structure of the very-low transconductance (VLT) OTA is presented in [52], and it results in suitable linearity.

Electrocardiogram (ECG) monitoring requires advanced wearable technologies that require sensors and integrated circuits. In [53], an on-chip integrated ECG signal acquisition system is presented where a OTA with a capacitive-resistive feedback network is employed. The signal-to-noise ratio differs from 35.7 dB to 38.6 dB. The experimental results reveal that the presented circuit can be used in long-term and wearable applications. The structure of the proposed design is provided in Figure 18.

Song et al. presented a low-voltage current-reuse chopper-stabilized front-end amplifier that can be used for fetal ECG monitoring [54]. In this design, the peak charge-pump efficiency is 90%, and the consumed power is 1.56 μW. The system’s concept is presented in Figure 19 with pop-outs to provide details.

For concurrently monitoring EEG signals, in [55] a wearable active concentric electrode was presented, with body-coupled communication (BCC)-based concurrent recording/transmitting (see Figure 20). The presented system achieves 100 dB CMRR with a power consumption of 7.4 μW.

In [56], multi-channel biosignal recording systems were recorded to tackle the problems of CMRR reduction due to the impedance mismatch at the input terminals. For this case, an equivalent electrical circuit model of the input interface is provided in the study.

For medical ultrasonic transmitter applications, in [57] a high-voltage linear PA was designed, simulated, and measured. In this design, to reinforce the signal linearity of the amplifier, the digital predistortion (DPD) technique is applied, including a 7-bit digital-to-analog converter (DAC), an analog-to-digital converter, and a digital field-programmable gate array (FPGA). This amplifier achieves a power efficiency of 30%. The structure of PA includes low-voltage amplifier (LV AMP) and high-voltage current-feedback amplifier (HV CFA) is depicted in [57], where the linear amplifier is presented with the help of DPD.

Early detection of diseases can enable treatment to the highest degree. Hence, in [58], a compact electronic module for counting the biological living cells was presented. This design helps in sensing the unbalanced impedance between the sensing microelectrodes. The general measurement setup is provided in [58], which includes the instrumentation amplifier stage for sensing the difference in signals and also a lock-in amplifier stage for demodulating the signals.

In another study, a chopper-stabilized operational amplifier with low noise was presented [59]. The total current consumption is 117.2 μA and the power supply is 1.8 V. A summary of the achieved results using this design is as follows: the DC gain is 132 dB, the CMRR is 192 dB, and the bandwidth is 2.69 MHz. The overall structure is presented in Figure 21.

For reducing the differential electrode offset, a low-power current feedback instrumentation amplifier is presented in [60]. The presented design is simulated in 180 nm CMOS technology, where the input noise is 3.8 μV${}_{rms}$ and the power consumption is 12.7 μW.

In [61], optimization methods for designing wireless medical implant devices was presented. This method leads to optimized spiral coil designs for achieving maximum power transfer. For feeding the transmitter coil, a class-E amplifier was designed.

## 4. Biomedical Amplifiers from IC and Layout Perspectives

When designing amplifiers with CMOS technology, the layout is provided before fabrication, and the layout must pass LVS and DRC specifications in the simulation tool [62,63,64,65]. This section is devoted to the presentation of some examples of generated layouts for biomedical amplifiers.

In [66], based on indirect current feedback, a fully-differential (FD) instrumentation amplifier was designed with 180 nm CMOS technology, where the supply voltage was 1.8 V. The presented bandwidth was 5.83 MHz and the DC current consumption was 266.4 µA. Figure 22 shows the layout of IC with an area of 0.54 mm${}^{2}$, and Figure 23 shows the proposed amplifier at the transistor level.

The integrated amplifier presented in [67] includes operational transconductance amplifier, chopper input, folded cascode, inverter-based input, multistage amplifier, and current reuse, and the general layout is presented in Figure 24.

## 5. Biomedical Applications from the Measurement Perspective

For any of the designs presented above, providing a suitable measurement setup is also an essential task. This section is devoted to the presentation of various aspects of medical measurements such as an integrated analog readout for multi-frequency bioimpedance measurements, an AC instrumentation amplifier for bioimpedance measurements, and an analog realization of a fractional-order skin-electrode model for tetrapolar bioimpedance measurements.

For the measurement aspect, Cheon et al. presented an effective impedance measurement feature for bioimpedance consideration, where it measures the magnitude and the real part of the complex impedance [68]. Figure 25 shows the presented electrochemical impedance spectroscopy system, which measures the impedance over the targeted frequency band. The consumed power is 513 µW, and the accuracy of measurement is improved by 30% in comparison to the conventional measurement methods.

In another study, an indirect current feedback method was presented for conducting bioimpedance measurements [69]. It uses degeneration resistors where the resistors are low. The design was simulated in 180 nm CMOS technology where the voltage gain is 4 *v*/*v* with 1.8 V supply voltage. The general structure of the presented indirect current feedback method is illustrated in Figure 26.

Another method for bioimpedance measurements was described in [70] and was simulated in 90 nm CMOS technology. The presented method is based on the fractional-order capacitor emulator that is shown in Figure 27. The simulation results demonstrated that the magnitude is low and the phase error is minimal; hence, the presented structure is suitable for bioimpedance measuring situations of up to 10 KHz.

Table 1, at the end of this section summarizes the various contributions and specifications of each reported work in this study.

## 6. Conclusions

In healthcare communication systems, providing high-performance biomedical devices is critical, as the recognition of any disease and its treatment depends greatly on the performance of these devices. Typically, the effectiveness of each medical device depends on the application of each one. Amplifiers are the most important part of medical devices, especially for the implanted devices. They are employed for various applications with vastly different topologies and structures. This survey is devoted to the summary of the designed amplifiers in very recently published literature aimed at usage in medical applications. By reading this review, researchers can obtain a general view of the various amplifier designs and can find a suitable design for their problems. In the future, the importance of emerging fifth-generation (5G) and next-generation technologies can be studied in biomedical amplifiers, because these technologies provide a platform for transferring data between the patient and medical doctors.

## Figures and Tables

**Figure 1 sensors-23-02277-f001:**
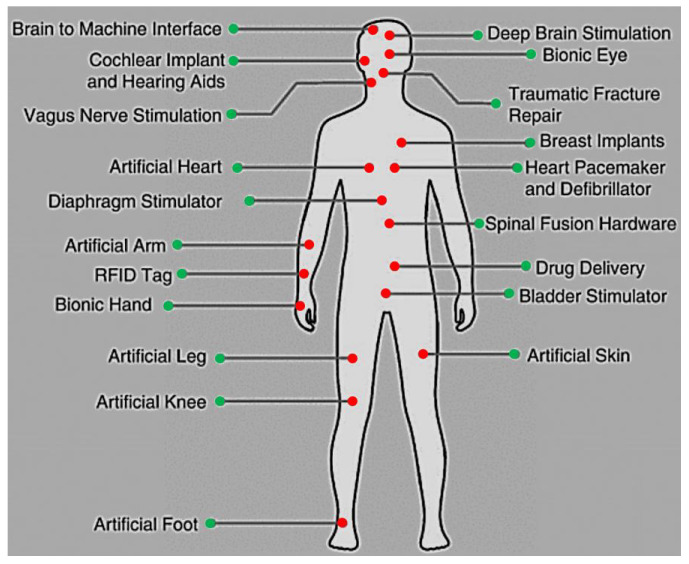
Diverse body locations of implanted devices.

**Figure 2 sensors-23-02277-f002:**
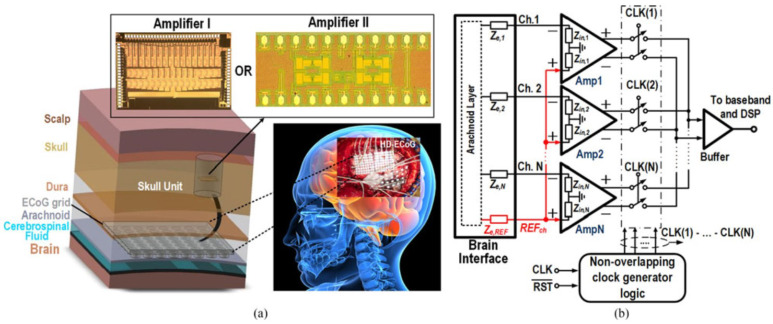
(**a**) Structure of brain signal acquisition (BSA) front-ends, (**b**) low-noise amplifier presented in Reprinted/adapted with permission from Ref. [21]: Copyright 2023, IEEE.

**Figure 3 sensors-23-02277-f003:**
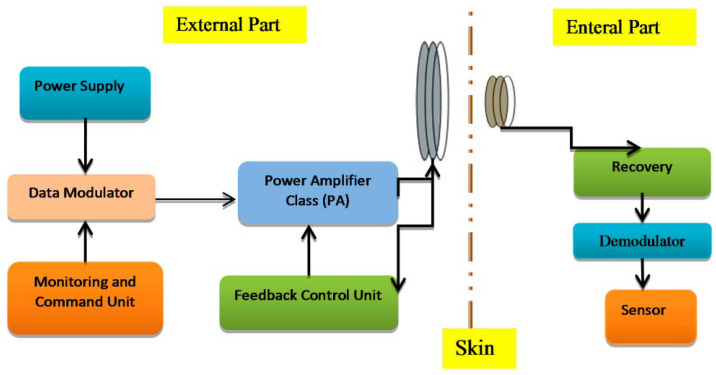
Power amplifier and its importance in implanted devices [22].

**Figure 4 sensors-23-02277-f004:**
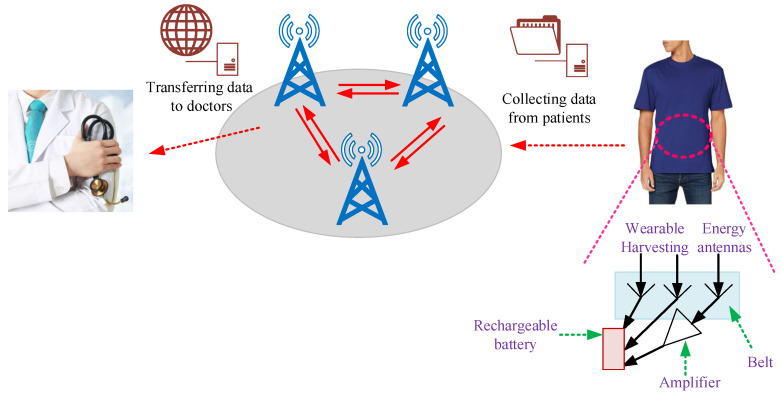
Methodology for transferring data from patient to medical doctor [23].

**Figure 5 sensors-23-02277-f005:**
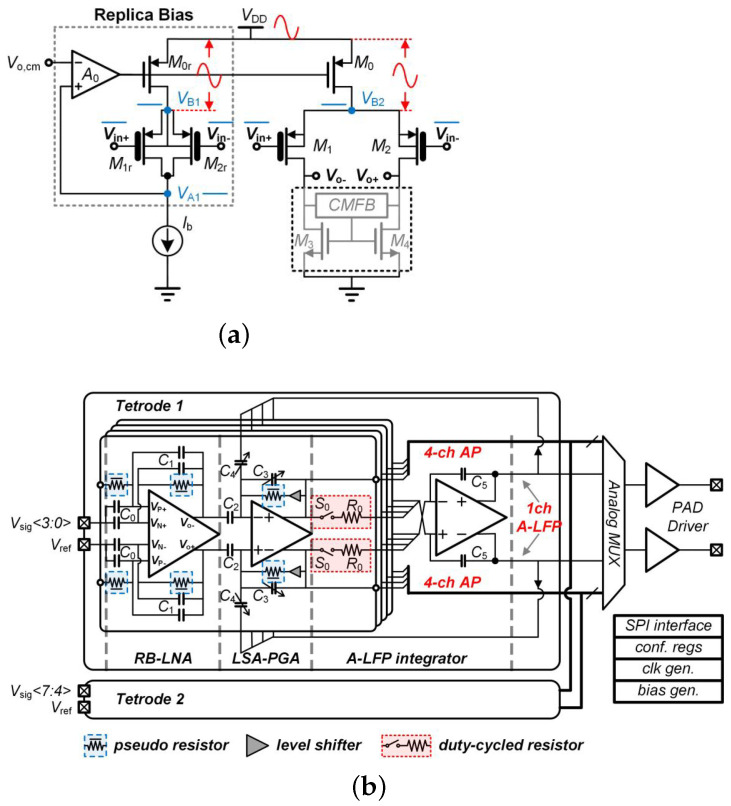
(**a**) Presented amplifier including the replica biasing; (**b**) the implementation of amplifier in the presented neural interface SoC. Reprinted/adapted with permission from Ref. [28]: Copyright 2023, IEEE.

**Figure 6 sensors-23-02277-f006:**
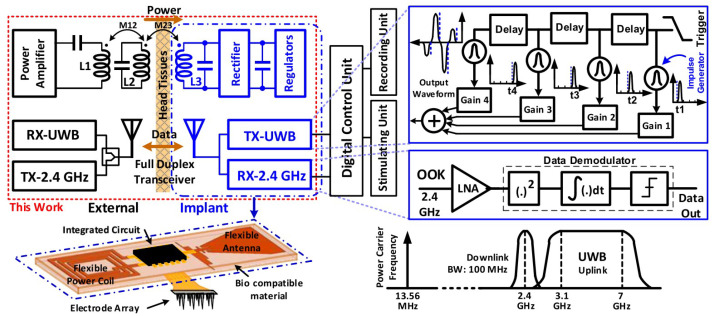
The implantable neural recording device. Reprinted/adapted with permission from Ref. [39]: Copyright 2023, IEEE.

**Figure 7 sensors-23-02277-f007:**
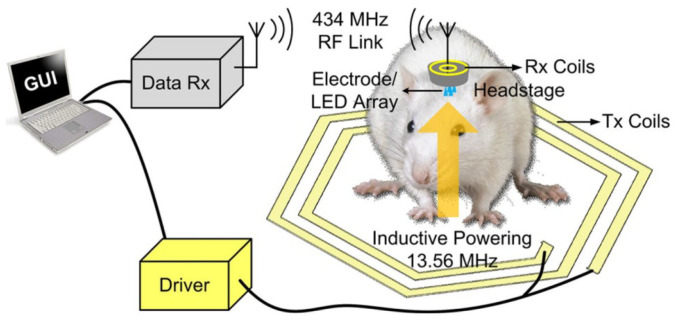
Practical implementation of SoC on a rat. Reprinted/adapted with permission from Ref. [40]: Copyright 2023, IEEE.

**Figure 8 sensors-23-02277-f008:**
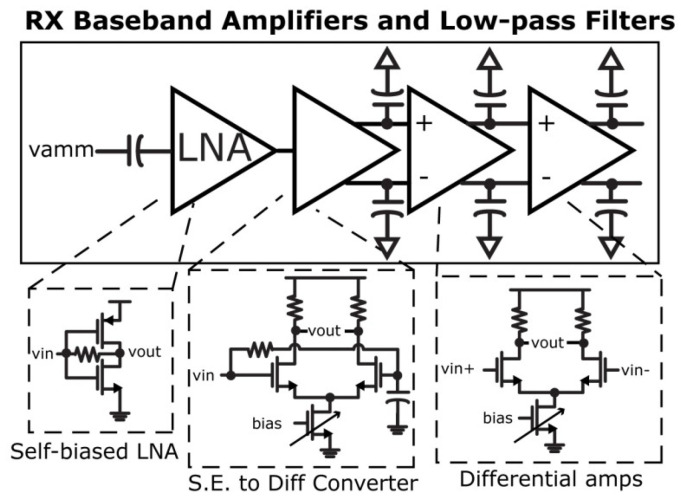
Designed amplifiers with filters in [42]: Reprinted/adapted with permission. Copyright 2023, IEEE.

**Figure 9 sensors-23-02277-f009:**
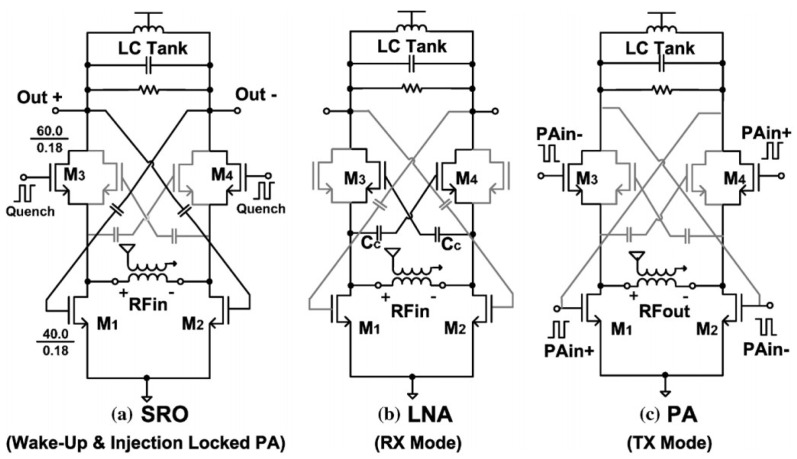
Detailed structures of SRO, LNA, and PA designs for the MICS transceiver. Reprinted/adapted with permission from Ref. [44]: Copyright 2023, IEEE.

**Figure 10 sensors-23-02277-f010:**
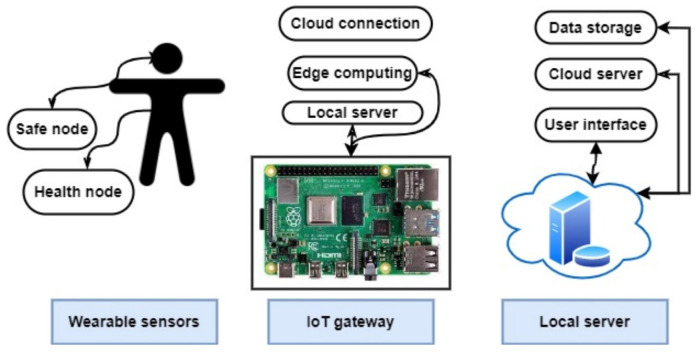
Presented healthcare-monitoring network in [46].

**Figure 11 sensors-23-02277-f011:**
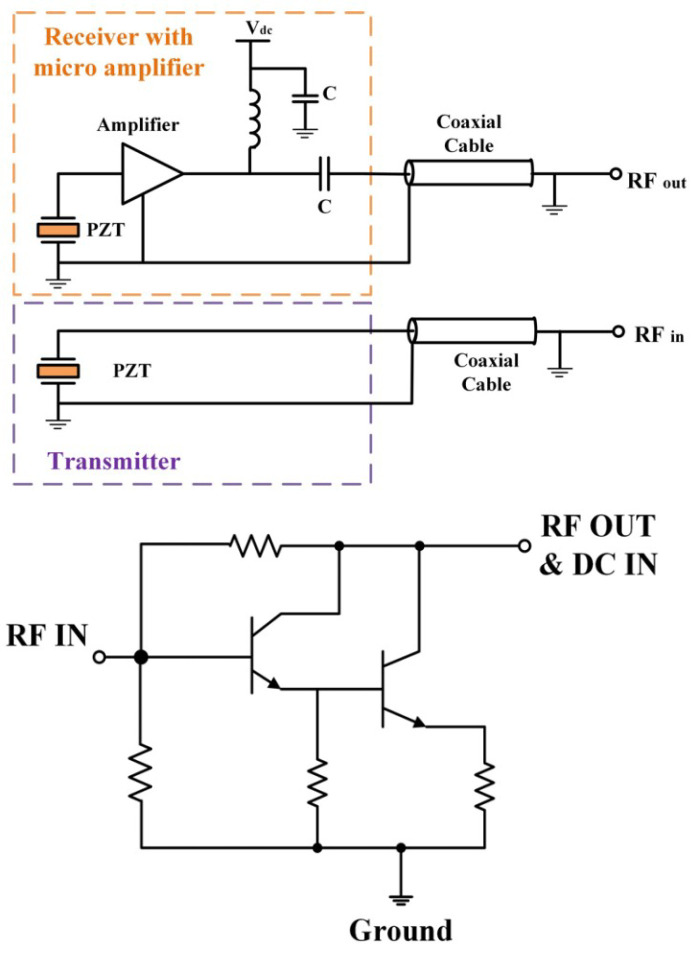
The general structure of proposed transducer (**top**) with configuration of proposed amplifier (**bottom**) [48].

**Figure 12 sensors-23-02277-f012:**
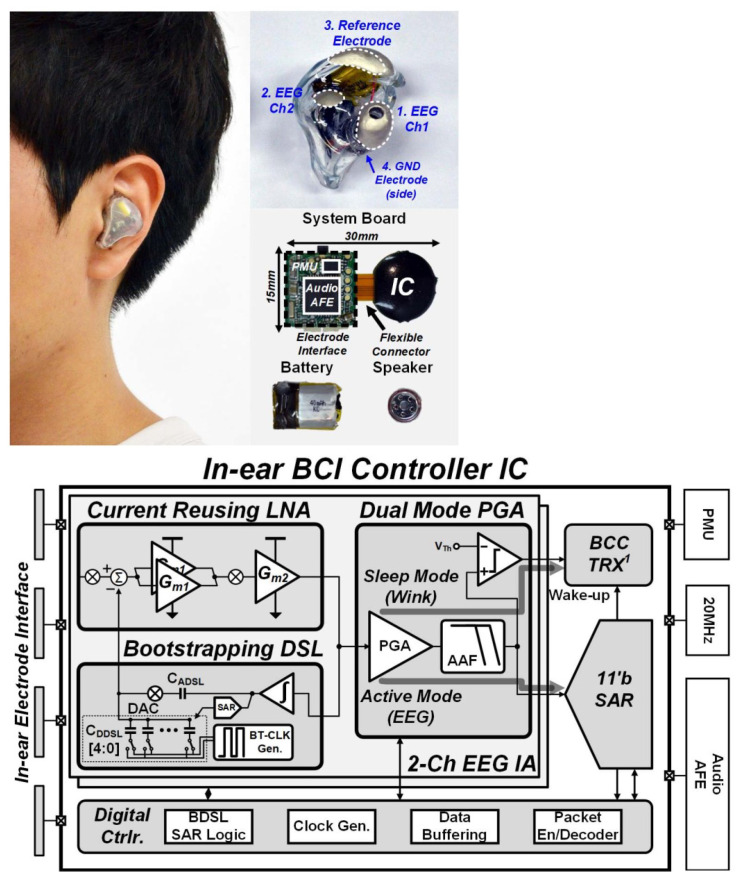
Practical implementation of IC in the ear of a human (**top**). Presented IC structure in detail (**bottom**). Reprinted/adapted with permission from Ref. [25]: Copyright 2023, IEEE.

**Figure 13 sensors-23-02277-f013:**
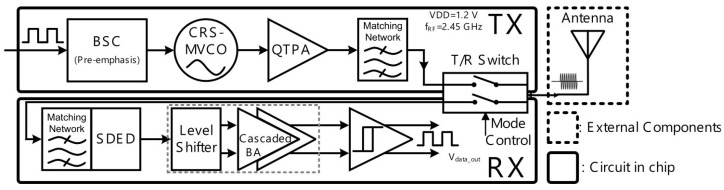
Presented radio frequency TRX in [16].

**Figure 14 sensors-23-02277-f014:**
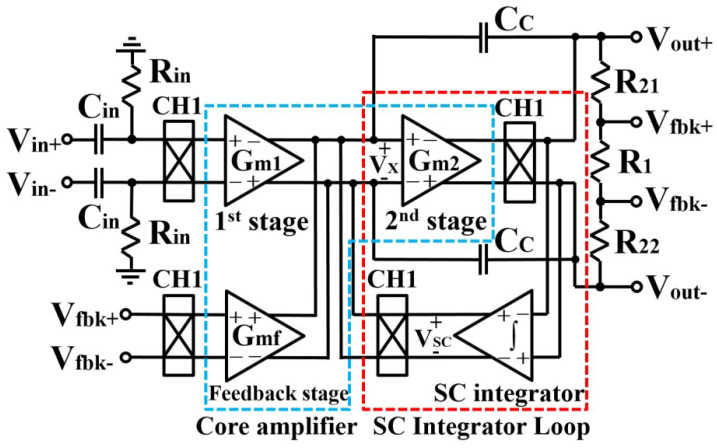
Presented current feedback instrumentation amplifier in [49].

**Figure 15 sensors-23-02277-f015:**
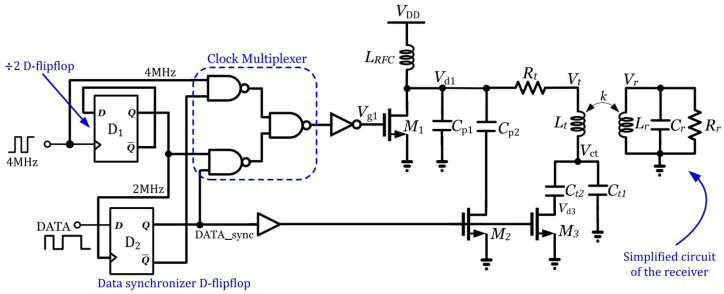
Structure of the Class-E PA with FSK data modulation presented in [34]. Reprinted/adapted with permission: Copyright 2023, IEEE.

**Figure 16 sensors-23-02277-f016:**
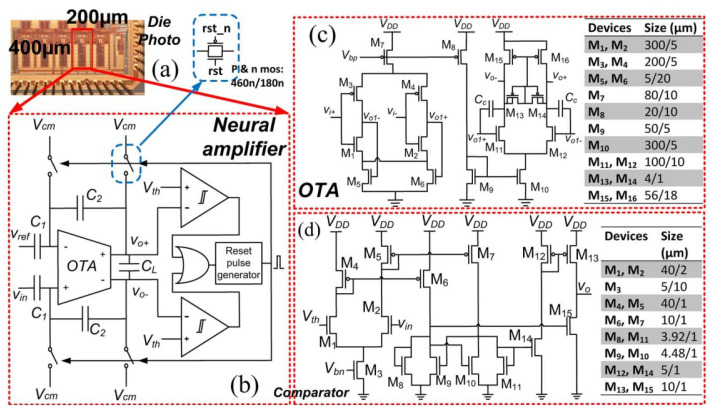
Presented power system in [50] to be used in neural recording: (**a**) Rectangle presents the area where the amplifier is fabricated, (**b**) Just one type of comparator is required, (**c**) OTA design, (**d**) Comparator design. Reprinted/adapted with permission. Copyright 2023, IEEE.

**Figure 17 sensors-23-02277-f017:**
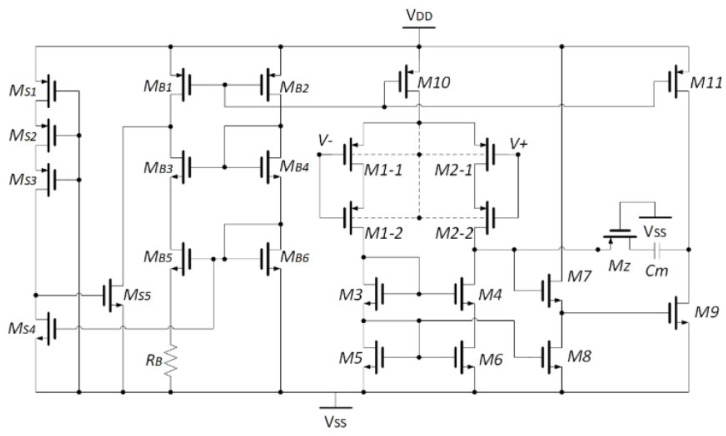
Presented OTA design in [51].

**Figure 18 sensors-23-02277-f018:**
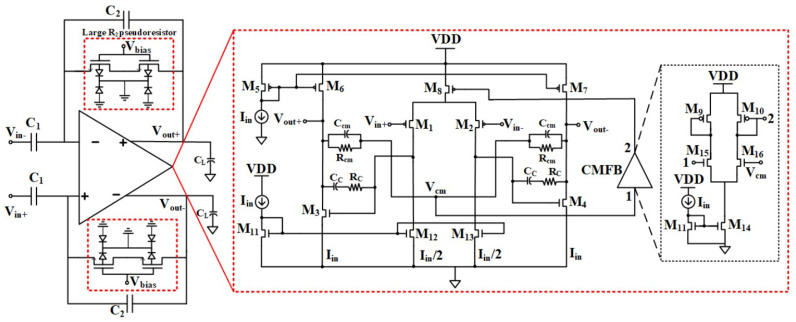
Presented OTA design in [53]. Reprinted/adapted with permission: Copyright 2023, IEEE.

**Figure 19 sensors-23-02277-f019:**
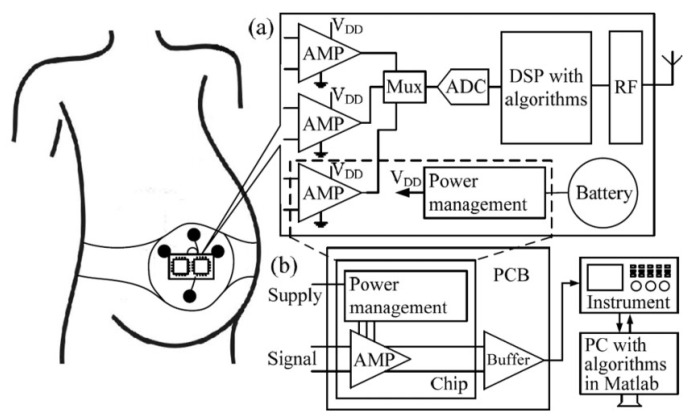
Fetal ECG monitoring; (**a**) the overall system scheme, (**b**) close-up of the measurement setup [54]: Reprinted/adapted with permission: Copyright 2023, IEEE.

**Figure 20 sensors-23-02277-f020:**
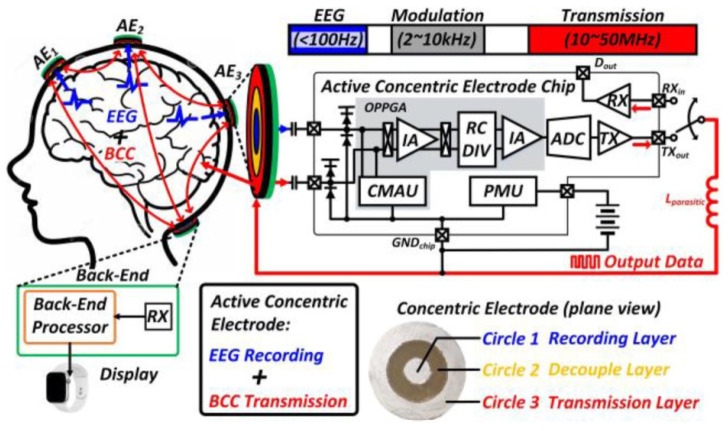
The overall configuration of a wearable active concentric electrode [55]. Reprinted/adapted with permission: Copyright 2023, IEEE.

**Figure 21 sensors-23-02277-f021:**
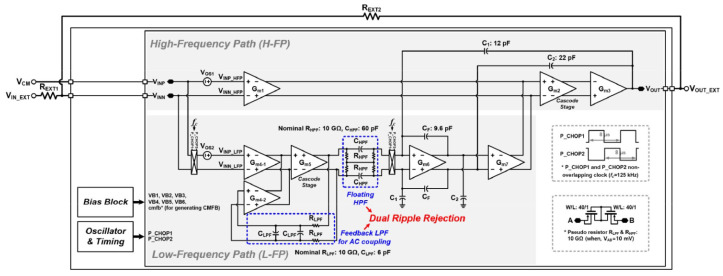
Structure of chopper-stabilized operational amplifier presented in [59]. Reprinted/adapted with permission: Copyright 2023, IEEE.

**Figure 22 sensors-23-02277-f022:**
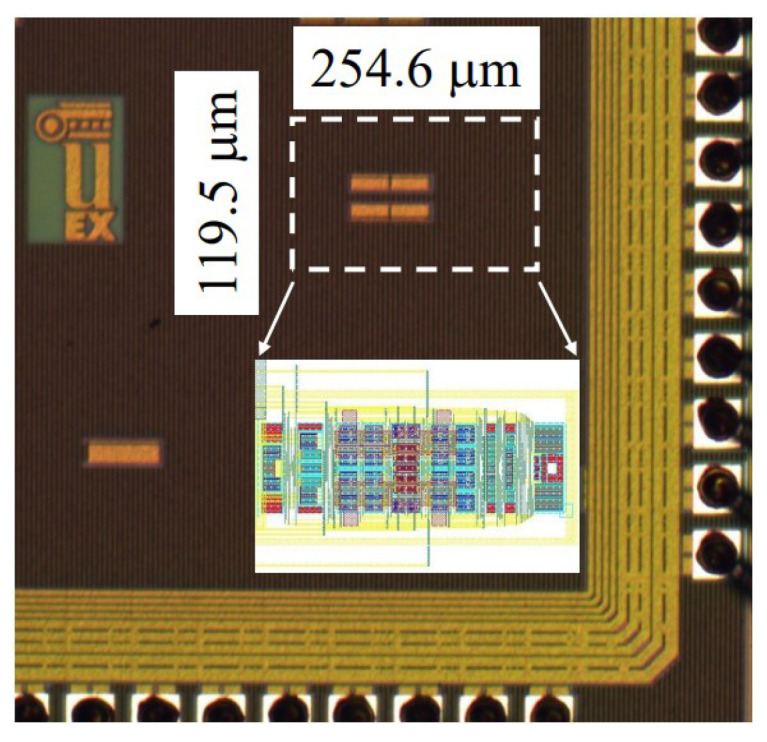
Layout of presented amplifier in [66].

**Figure 23 sensors-23-02277-f023:**
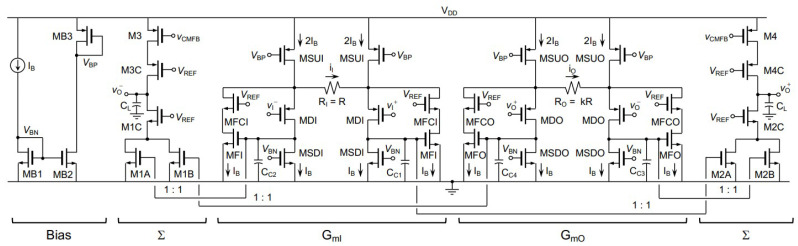
Circuit design of generated layout in Figure 22 where the design is captured from [66].

**Figure 24 sensors-23-02277-f024:**
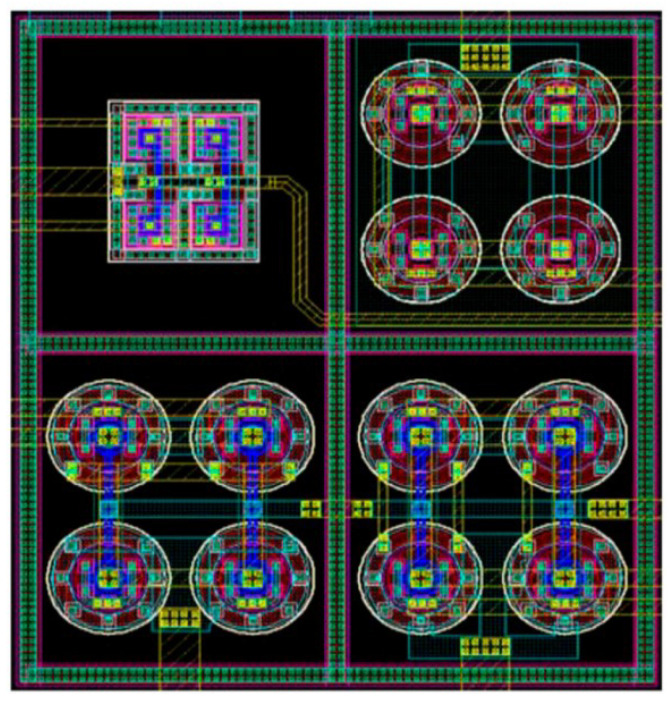
Layout of amplifier IC presented in [67].

**Figure 25 sensors-23-02277-f025:**
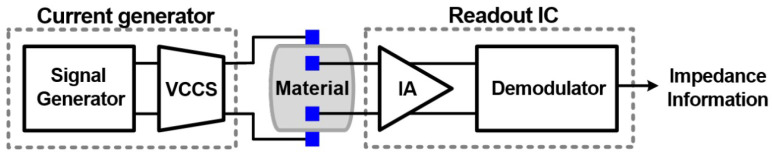
General structure of electrochemical impedance spectroscopy presented in [68] with a combination of voltage-controlled current source (VCCS), integrated circuit (IC), and instrumentation amplifier (IA).

**Figure 26 sensors-23-02277-f026:**
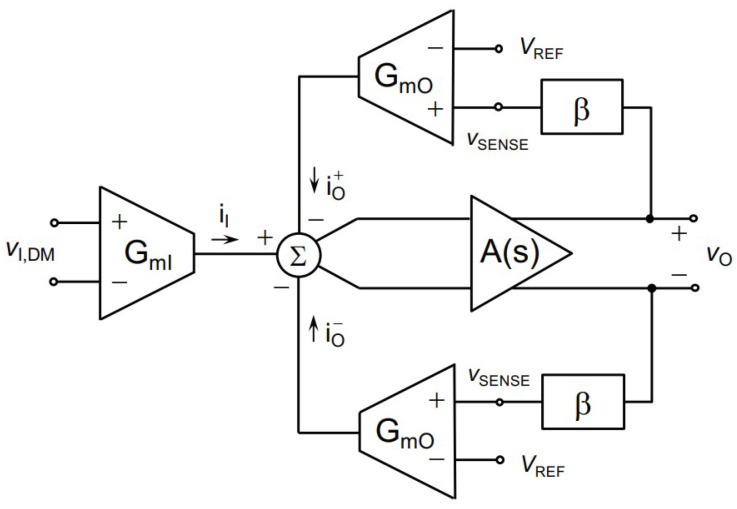
General structure of indirect current feedback presented in [69].

**Figure 27 sensors-23-02277-f027:**
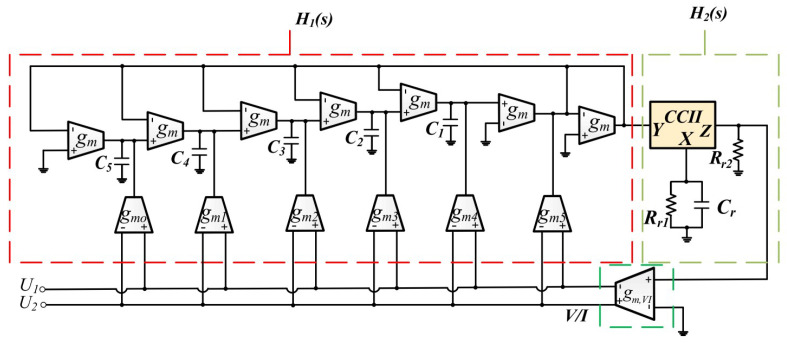
The general design of fractional-order capacitor emulator presented in [70].

**Table 1 sensors-23-02277-t001:** Summary of some reported biomedical amplifiers with the focuses and output specifications of each one.

Ref.	Scope	Contribution	Specifications
[1]	Provides wearable systems for monitoring health status.	Sensing the vital biosignals through a low-noise, energy efficient readout front-end.	Meeting the requirements of gain 45.94 dB, input referred noise of 0.184 µV${}_{rms}$, consumed power of 145.9 nW, and supply power of 1 V.
[21]	Provides two brain signal acquisition front-ends.	Providing low-noise amplifier arrays operating in a MOSFET weak inversion region.	Meeting the requirements of 0.216 µW with 0.4 V supply where the design is simulated in 180 nm CMOS technology.
[28]	Provides an eight-channel energy-efficient analog front-end for neural recording.	Improvements in power supply rejection ratio and dynamic range.	Meeting the requirements of 340 nW consumed power, 0.7 V supply voltage, input-referred noise is 6.7 µV, and the simulated environment is 65 nm CMOS technology.
[39]	Provides a fully-integrated low-power full-duplex transceiver.	Supporting high-density and bidirectional neural interfacing applications.	Meeting the requirements of 41.6% efficiency with power consumption of 10.4 mW when the design is simulated in 180 nm CMOS technology.
[40]	Provides a wireless and battery-less trimodal neural interface system-on-chip.	Supporting 16-channel neural recording, 8-channel electrical stimulation, and 16-channel optical stimulation.	Meeting the requirements of 55–70 dB for analog front-end with low/high cut-off frequencies of 1–100 Hz/10 kHz.
[42]	Provides brain–machine interface system.	Supporting Class-E/Fodd amplifier with a current-sense resistor.	Meeting the requirements of 54% efficiency at 309 MHz.
[44]	Provides a low-power transceiver for medical implant communication systems.	Includes wake-up signal reception, data-link binary frequency-shift keying reception, and transmission.	Meeting the requirements of a 97 dBm sensitivity with 2 mW power consumption that is simulated in 180 nm CMOS technology.
[44]	Provides a low-power transceiver for medical implant communication systems.	Includes wake-up signal reception, data-link binary frequency-shift keying reception, and transmission.	Meeting the requirements of a 97 dBm sensitivity with 2 mW power consumption that is simulated in 180 nm CMOS technology.
[48]	Provides an endoscopic transducer and is applied in medical ultrasound imaging.	Limiting endoscopic ultrasound detection depth.	Meeting the requirements of a 20 MHz center frequency.
[25]	Provides an in-ear brain–computer interface (BCI) controller system.	Includes a system-on-chip with electroencephalography readout and body channel communication.	Meeting the requirements of 82.9 µW power with 84% average accuracy.
[16]	Provides a cross-domain integration system.	Includes a gene transfer technique with a pre-clinical trial, an on-chip circuit design, an off-chip hardware with peripheral unit integration, and custom software.	Meeting the requirements of intermodulation distortion with a third order of approximately 68 dB with 16-bit output data.
[49]	Implements an amplifier for bio-potential signal acquisition applications.	Includes low-noise, low-power, chopper-stabilized, current-feedback specifications.	Meeting the requirements of input referred noise voltage of 0.75 µV at frequency band of 0.01–100 Hz, power dissipation of 2.3 µW, and a common-mode rejection ratio of 125 dB.
[34]	Implements Class-E amplifier for transferring data to medical implants.	Includes frequency-shift keying as a data modulation scheme.	Meeting the requirements of a power transfer efficiency of 25% and a power delivered to the load of 126 mW.
[50]	Presents a signal folding and reconstruction scheme for neural recording.	Includes 1/f${}^{n}$ characteristics of neural signals.	Achieves a gain of 54.2 dB, bandwidth of 5.7 kHz, and power dissipation of 2.52 µW.
[51]	Presents a front-end amplifier.	Includes 1/f${}^{n}$ characteristics of neural signals.	Achieves a power consumption of 7.6 µW with common-mode rejection ratio of 61 dB and the simulation platform is 180 nm CMOS technology.
[53]	Presents an electrocardiogram monitoring system.	Includes an on-chip integrated electrocardiogram signal acquisition system along with dry electrodes for wearable and long-term monitoring.	Achieves a gain of 43.09 dB and input-referred noise of 2.81 µV${}_{rms}$.
[54]	Presents a low-voltage current-reuse chopper-stabilized frontend amplifier.	Results in individual tuning of the noise in each measurement channel and minimizes the total power consumption.	Achieves a noise level of 0.34 µV${}_{rms}$ with a power consumption of 1.17 µW.
[59]	Presents a chopper-stabilized low-noise multipath operational amplifier with dual ripple rejection loops.	Results in lower noise and a wider bandwidth.	Achieves total current consumption of 117.2 µA with 1.8 power supply, where the design is simulated in 180 nm CMOS technology.

## Data Availability

Not applicable.

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
