# Peer review of "Amplifiers in Biomedical Engineering: A Review from Application Perspectives"

_sensors, 2023, doi:10.3390/s23042277_

Round 1

Reviewer 1 Report

The manuscript titled: Amplifiers in Biomedical Engineering: A Review from Application Perspectives. This review devotes to collect and provide a comprehensive review of the various designed implanted amplifiers for advanced biomedical applications. The manuscript is interesting and worth consideration for publication.

Author Response

Dear Reviewer,

in the attached file you can find our reposnses.

Thank you.

Reviewer 2 Report

Dear Authors,

I think it is of an author's best interest to have a review with the highest amount of fair-criticism as possible, thus having his/her name associated with high-quality work. Minding the time constraints to review this paper, I spent the maximum amount of time I could on it and tried to be as critical as I could.

As a Reviewer and Analog/RF IC Design Engineer, my general opinion is that the topic is interesting.

However, I think the manuscript is not ready yet for publication, it still needs maturing in terms of the accuracy in theory and also in formatting.

Next, see my complete review, minor and major issues are blended. The comments follow the same order as the paper.

1) I think that your Abstract is well-written. But you should be more specific regarding the methods and the implementations which are tested and proposed in your work. As I can observe, you provide more general information in abstract, please be more specific. Thank you.

2) Despite of the large number of works reported, cited, this work lack of a deeper level of understanding and criticism that a review paper should show. Please add extra information for each work in which you explain more the main design methodology and design concept.

3) Moreover, provide Tables with related metrics and information related to design procedure. Only specific metrics because as you refer this review paper is application perspective.

4)Biomedical applications are related also to measurements. As I can observe you have missed citations related to this concept. For example you can check the following:
a)An integrated analog readout for multi-frequency bioimpedance measurements

b)AC instrumentation amplifier for bioimpedance measurements

c)Analog realization of fractional-order skin-electrode model for tetrapolar bio-impedance measurements.

etc

Please check related work.

5) Can you please provide a sub-section related to IC Design and one for discrete?

6) In Section 2 please provide an extra figure in which you include the main design concept which biomedical applications follow. It will upgrade your work.

7) As you refered to your conclusion, "Any researcher 309 by reading this review, can get a fast view on the various amplifier designs and can find 310 the most suitable design for their problems." This statement is not 100% true, since you do not provide the appropriate metrics. The IC design needs both metrics and design difficulties in order to be 100% accurate. Please update your work accordingly.

8) Create sub-sections or sub-categories for each specific case. It will upgrade your work and help the reader to find out each category.

Author Response

(The authors gave the same response as above.)

Reviewer 3 Report

This research focuses on gathering and providing an in-depth analysis of the many implanted amplifiers built for cutting-edge biomedical applications.

It will be good, if the author summarises the merits and demerits of various amplifier designs an a tabular form and its parameters in a graphical form. 

Poor resolution for the figures Figure 1, Figure 3, Figure 9 and Figure 12.

The conclusion of the manuscript is too short. It may also give suggestions for future study and identify gaps that need to be filled.

Author Response

(The authors gave the same response as above.)

Round 2

Reviewer 2 Report

Dear Authors,

My firm belief is that your work is ready for publication. You have deal with all my concerns.